# Gender-Related Differences in Flood Risk Perception and Behaviours among Private Groundwater Users in the Republic of Ireland

**DOI:** 10.3390/ijerph17062072

**Published:** 2020-03-20

**Authors:** Cillian P. McDowell, Luisa Andrade, Eoin O’Neill, Kevin O’Malley, Jean O’Dwyer, Paul D. Hynds

**Affiliations:** 1The Irish Longitudinal Study on Ageing, Trinity College Dublin, Dublin 2, Ireland; Cillian.McDowell@tcd.ie; 2School of Biological, Earth and Environmental Sciences, Distillery Fields, University College Cork, Cork T12 YN60, Ireland; luisa.andrade@ucc.ie (L.A.); jean.odwyer@ucc.ie (J.O.);; 3Irish Centre for Research in Applied Geosciences (iCRAG), University College Cork, Cork T12 YN60, Ireland; 4UCD Environmental Policy, University College Dublin, Dublin 4, Ireland; eoin.oneill@ucd.ie; 5UCD Earth Institute, University College Dublin, Dublin 4, Ireland; 6Department of Psychology, University of Limerick, Limerick V94 T9PX, Ireland; kevin.omalley@ul.ie; 7Environmental Research Institute, University College Cork, Cork T12 YN60, Ireland; 8Environmental Sustainability & Health Institute, Technological University Dublin, Dublin 7, Ireland

**Keywords:** flood risk, perception, gender, private groundwater, behaviour, psychology

## Abstract

Extreme weather events including flooding can have severe personal, infrastructural, and economic consequences, with recent evidence pointing to surface flooding as a pathway for the microbial contamination of private groundwater supplies. There is a pressing need for increasingly focused information and awareness campaigns to highlight the risks posed by extreme weather events and appropriate subsequent post-event actions. To date, little is known about the presence, directionality or magnitude of gender-related differences regarding flood risk awareness and behaviour among private groundwater users, a particularly susceptible sub-population due to an overarching paucity of infrastructural regulation across many regions. The current study investigated gender-related differences in flood risk perception and associated mitigation behaviours via a cross-sectional, national survey of 405 (168 female, 237 male) private groundwater supply users. The developed survey instrument assessed socio-demographic profile, previous flood experience, experiential and conjectural health behaviours (contingent on previous experience), and Risk, Attitude, Norms, Ability, Self-regulation (RANAS) framework questions. Statistically significant gender differences were found between both ‘Norm—Descriptive’ and ‘Ability—Self-efficacy’ RANAS elements (*p* < 0.05). Female respondents reported a lower level of awareness of the need for post-flood action(s) (8.9% vs. 16.5%), alongside a perceived “lack of information” as a reason for not testing their domestic well (4.9% vs. 11.5%). Conversely, male respondents were more likely to report awareness of their well location in relation to possible contamination sources (96.6% vs. 89.9%) and awareness of previous water testing results (98.9% vs. 93.0%). Gender-related gaps exist within the studied private groundwater reliant cohort, a sub-population which has to date remained under-studied within the context of climate change and extreme weather events. Accordingly, findings suggest that gender-focused communication and education may represent an effective tool for protecting current and future generations of global groundwater users.

## 1. Introduction

A recent global review reports that surface flooding may result in the contamination of groundwater systems via direct source ingress and/or preferential/bypass flows, thereby leading to waterborne enteric infections among private groundwater supply users [1]. Recent projections indicate significant global increases in the frequency and intensity of flooding events due to climate change [2,3,4], with the Republic of Ireland (RoI) forecast as one of the most severely affected countries in Europe relative to the mean population proportion residing in flood-prone areas [5]. Private groundwater systems account for a significant proportion of daily water for human consumption in many regions. For example, approximately 12% (4.4 million) and 14.7% (48 million) of the Canadian and U.S. populations, respectively, are reliant on these systems for daily consumption, compared with ≈750,000 Irish residents (16% of the national population) [6,7]. One of the defining characteristics of private groundwater systems in many regions is a lack of overarching regulation with respect to source location, design, and maintenance. Accordingly, it is critical that private groundwater users are aware of the risks posed by flooding to their domestic supplies and the necessary actions to take following localised flooding, as municipal/governmental support and expertise are frequently unavailable. However, evidence suggests that appropriate actions, such as well water testing, are not commonly undertaken post-flooding for a variety of reasons, including optimism bias regarding both the susceptibility to flooding and possible post-incident contamination, as well as a lack of information on the risks of flooding and appropriate mitigation responses [8]. Given the public health impacts of climate change exacerbation, evidence-based, appropriately framed strategies promoting an increased awareness of potential contamination mechanisms are necessary to motivate precautionary behavior [1].

Over the past three decades, the multiple social, economic, demographic, and geographical factors influencing people’s perceptions and responses to risk in numerous domains including personal health and wellbeing, environment, education and finance has received increasing attention [9,10]. Indeed, risk related research has deviated from the disciplines of statistics and economics and has shifted towards social and behavioural psychology. Psychological studies have demonstrated a subjective dimension to risk perception, with sociologists suggesting that risks are socially and culturally constructed and thus, may exhibit a marked gender dimension [11]. As such, it is now accepted that risk perception may differ across genders, albeit moderated by other structural factors (e.g., class, ethnicity). Studies suggest that differences in gender play a significant role in subjective risk assessment [11]; within an environmental context, gender has received specific attention with empirical research indicating a reasonably consistent gap between male and female cohorts. Bord and O’Connor (1997) report that female study participants demonstrated a higher level of concern for environmental and hydrological threats under certain conditions [12], while Hunter et al. (2004) similarly found that females engaged in more environmental behaviours including recycling, purchasing organic fruit/vegetables, and the minimisation of driving for environmental reasons [13]. However, these findings were predicated on study cohorts appropriately perceiving the risk associated with a specific domain [12]. Accordingly, understanding whether private well users exhibit a marked gender difference in their perception of post-flood risks and thus the importance of appropriate mitigation behaviours is essential to optimise the adoption of effective strategies. 

Historically, gender has been an understudied construct within health and medical research [14], with the issue only recently addressed within associated literature [15]. As such, it is critical that any potential gender differences within an environmental health context are assessed [16] as it has been noted that gender represents a vital component of health research [17]. Indeed, understanding gender issues and informing gender sensitive interventions is of particular relevance to permit the development and delivery of optimised human health interventions [18]. Within the context of flooding, gender differences have been reported within the literature, for example, several studies have found that female respondents tend to perceive the risk of floods more acutely than their male counterparts [19,20], and thus, may represent a specific target audience for risk reduction strategies. A study of flood-risk perception in the Republic of Ireland found gender differences in relation to the affective component of flood risk perception i.e., females were more likely to worry about natural hazards than males [21]. However, O’Neill et al. (2016) also showed that elevated risk perception did not translate into higher levels of protective behavior [21], perhaps highlighting that in spite of increased perception among women, traditional ‘gender roles’ (i.e., men as ‘protectors’) prevail and risk reduction measures are influenced more routinely by male perception. Similarly, the probability of purchasing flood insurance has been reported as being comparable (with male respondents) or lower among female respondents [20,22]. Conversely, Zaalberg et al. (2009) found no association between gender and the intention to undertake adaptive actions for flood damage minimisation [23] thus highlighting the inherent complexity of the issue. Notably, the abovementioned studies focused on “pre-flood” perceptions and behaviours; to date, few studies have explored gender related differences on post-flood risk perceptions and behaviours, and particularly as they relate to human health. This knowledge gap represents a key impediment to the development of “fit for purpose” risk reduction, as highlighted in the Hyogo Framework for Action, 2005–2015 [24] and as included in the post-2015 framework for disaster risk reduction, which calls for a gender perspective to be integrated into all decision-making processes. Moreover, as set out in the Sendai Framework for Disaster Risk Management 2015–2030, on which the Republic of Ireland is a signatory, *“Disaster risk reduction requires an all-of-society engagement and partnership. It also requires empowerment and inclusive, accessible and non-discriminatory participation, paying special attention to people disproportionately affected by disasters … A gender … perspective should be integrated in all policies and practices, and female leadership should be promoted* [25]. Similarly, the World Health Organisation distributed questionnaires among EU Member States and found that while most Member States had flood management plans in place, these did not generally address the needs of vulnerable groups or gender considerations [26].

The Republic of Ireland, which is characterised by an historic risk of flooding in concurrence with a relatively high level of private (unregulated) groundwater reliance (estimated at over 16% of the population) [6,7], serves as an optimal experimental site for the current study. From the mid-19th century, public policies concerning flooding as it relates to drainage for land improvement for agriculture have been introduced; urban flood events in the 1980s and 1990s saw a policy shift with a focus on protecting urban conurbations and necessary infrastructure from flooding. More recently, there has been a return to wider river-basin concerns and implementation of risk-based models to manage flood risks [27]. However, whilst increasingly holistic risk-based models are being pursued, there has been limited consideration of the link between flooding, contamination and human health [1]. This is despite Irish private household wells being identified as the likely source of serious health issues such as verotoxigenic Escherichia coli (VTEC) infections, for which the RoI has the highest incidence rates in Europe [28,29]. Moreover, recent flooding events have had extensive negative effects [30] with recent regional climate change projections predicting the scenario to worsen in the next 40 years [31,32]. As such, the Republic of Ireland is a highly pertinent case study to assess the gender-related differences in flood risk perceptions and post-flood mitigation behaviours among private groundwater supply users and is thus, the focus of this study.

## 2. Methods

### 2.1. Study Design

A cross-sectional online questionnaire was developed and undertaken with private groundwater users between November 2017 and February 2018. The questionnaire was hosted on SurveyMonkey and distributed among private well users aged ≥18 years residing in the Republic of Ireland via several non-expert interest groups (e.g., farmers organisations, countrywomen’s associations, etc.). The questionnaire focused on four main themes: (1) socio-demographic characteristics, (2) flood experience, (3) experiential and conjectural responses to flooding (i.e., health behaviours taken by those who have experienced floods near their groundwater supply versus intended health behaviours by those who have not), and (4) Risk, Attitude, Norms, Ability, Self-regulation (RANAS) [33] framework questions (Table 1 and Table 2). For the purpose of this study, the RANAS model has been employed as it has been shown to have high predictive accuracy of conjectural and experiential health behaviours following flooding [8]. The University College Dublin Human Research Ethics Committee granted approval for the overarching study (Ref: HS-17-47-deAndrade-O).

### 2.2. RANAS Framework

The RANAS human cognitive model comprises 16 behavioural factors that inform whether a “health behaviour” (e.g., well testing following a flooding event) is undertaken, with the model typically applied to inform and evaluate anticipated behaviour changes subsequent to an intervention. Most extreme weather events, including significant flooding, are inherently sporadic, and as such their specific timing and consequences are unpredictable. Therefore five RANAS components (maintenance self-efficacy, recovery self-efficacy, action planning, coping planning, and remembering) were not considered concomitant with overarching study aims; i.e., to explore gender-related differences in flood risk perception and post-flood mitigation behaviours. Accordingly, the RANAS model used for the current study represents an adapted version, as these elements were excluded. A description of the eleven RANAS factors included in the study questionnaire is provided in Table 1. For the purposes of surveying, respondents were asked to report agreement with these RANAS statements on a ranked Likert scale from 1 (strongly disagree) to 5 (strongly agree).

### 2.3. Statistical Analyses

One-way ANOVA was used to test for between-gender differences in response to each RANAS component, with Hedges’ *d* effect sizes (95% CI) used to quantify the magnitude of identified differences. Hedges’ *d* is adjusted to correct for the upward bias shown by Hedges’ *g* and Cohen’s *d* when sample sizes are relatively small [34]. Chi-square tests with Bonferroni correction (multiple (>2 rows) comparisons) were used to examine gender differences with respect to mitigative knowledge including treatment awareness, water treatment, prior water testing experience, compliance with Environmental Protection Agency testing guidelines (once/annum), and primary post-flood actions. Binary logistic regression was used to calculate odds ratios (ORs) between surveyed RANAS components, gender (and their interaction), and participants’ self-reported conjectural (i.e., no flood experience) and experiential (previous flood experience) behaviour, with flood experience defined as “direct” (i.e., those who had personally experienced flooding) or “indirect” (i.e., those who knew people who had experienced flooding). A significant interaction between a RANAS component and gender indicates that the relationship between the RANAS factor and the outcome (i.e., self-reported behaviour following the flooding events) was predicated on participant gender. Chi-square tests were further employed to examine gender differences between self-reported conjectural and experiential behaviours. Twenty-six respondents who did not report their level of flooding experience were excluded from these analyses. All statistical analyses were conducted using SPSS Version 26.0 (Armonk, NY, USA: IBM Corp.).

## 3. Results

### 3.1. Respondent Characteristics

Private water supply users (*n* = 405) representing all 26 administrative counties in the Republic of Ireland completed the survey (Figure 1). In all, 58.5% of respondents were male, with 16.8%, 40.5% and 42.7% falling within the ≤35 years, 35 to 49 years, and >49 years age ranges, respectively. Approximately three quarters of respondents (72.8%) had completed third-level education (i.e., post-secondary), while a significant majority (88.6%) owned their current residence. Respondents’ residences were situated in rural agricultural settlements (42%), rural non-agricultural settlements (52.8%) and small villages, towns or other (peri)urban settlements (5.2%). The survey cohort was served by both private household wells (81.7%) and private group water schemes (18.3%). One fifth (19.7%) of respondents reported direct experience of flooding adjacent to (within 100 m) their domestic groundwater supply.

### 3.2. Statistical Analyses

Female respondents reported feeling less equipped to get their well water tested if required (*p* = 0.030; *d* = −0.22, 95% CI: −0.42, −0.02) and were less likely to believe that people they knew would test their well water if flooding occurred near their well (*p* = 0.045; *d* = −0.20, 95% CI: −0.40, 0.00) (Table 3). As shown (Table 4), compared to male respondents, females were more likely to report a lack of awareness of the need for post-flood action (16.5% vs. 8.9%), more likely to report a lack of information on testing as a reason for not testing their well (11.5% vs. 4.9%), less likely to report awareness of their well location in relation to possible contamination sources (89.9% vs. 96.6%), and less likely to report an awareness of the results of previous water tests on their private wells (93.0% vs. 98.9%). Males were more likely to report not testing their well if previous test results were normal (26.2% vs. 12.2%; all *p* < 0.05; Table 4).

Binary logistic regressions indicated no significant interaction between RANAS factors and gender in predicting experiential (Table 5) or conjectural (Table 6 and Table 7) health behaviours following flooding (all *p* > 0.05), meaning that the relation between each RANAS factor and participants’ self-reported experiential or conjectural behaviours following flooding events was not dependent on participant gender. Three of the eleven tested RANAS factors, Perceived Vulnerability, Norm Descriptive, and Commitment significantly predicted experiential health behaviours (all *p* < 0.05; Table 5) among people with direct flood experience, albeit as mentioned, these were not predicated on respondent gender. Two of eleven RANAS factors (Personal and Commitment) significantly predicted conjectural health behaviours (all *p* < 0.05; Table 6) among people with indirect flooding experience. Nine of eleven RANAS factors (not including Severity and Injunctive) significantly predicted conjectural health behaviours (all *p* < 0.05; Table 7) among people with no flooding experience. There were no gender differences with respect to experiential and conjectural testing behaviours within people with direct, indirect, or indirect flooding experience (Table 8).

## 4. Discussion

This study investigated the presence of gender differences as they relate to flood risk perception and post-flood mitigation behaviours among 405 private groundwater supply users in the Republic of Ireland. Specifically, this study sought to examine whether the predictive patterns put forward by the RANAS model [33] operate differently for males and females in the context of post-flood health behaviours such as groundwater testing. Previous research has reported that the female subpopulation will be more affected by climate change than their male counterparts, while conversely men typically pollute more than women [35,36]. Thus, a critical inequity exists, which to date, has not been examined or addressed among communities or regions reliant on private groundwater systems, primarily located in rural areas. This is particularly significant as men and women have been shown to interact with water resources and landscapes in different ways, with little research undertaken to address this issue [37]. Previous research has identified a gap between men and women in the context of risk perception such that, when aware of the risk, women tend to perceive environmental and hydrological risks more acutely than men [20,38], although this is not always the case [39]. In general, women appear to take increased preventative action to mitigate against risk, however, this propensity towards protection does not always appear to translate into the domain of flood-related behaviours. Bradford et al. (2012) and Scolobig et al. (2012) for example, both find males reporting higher flood preparedness levels than females [40,41]. As such, it was expected that there would be some degree of difference between both the perception of flood risk and the performance of post-flood health behaviours between the sexes. 

It is important to note that according to the most recent Irish census (2016), approximately 50.4% of the national population identifies as female [42], and thus, the current study sample does not strictly reflect the gender distribution within the Republic of Ireland (i.e., female sub-sample, 41.5%). Previous studies of environmental issues within rural areas, and particularly those focusing on water resources and waste management have reported similar difficulties in recruiting gender-balanced survey cohorts [43,44]. Flanagan et al. (2015) previously surveyed 525 private groundwater-reliant householders across central Maine (U.S.) regarding well testing and treatment practices in an area characterised by high geogenic groundwater arsenic [45]. Although the male/female ratio within the background population was 49.4%/50.6%, the surveyed sample displayed a slight gender skew of 54%/46%. Similarly, Hynds et al. (2018) surveyed 533 septic tank users from rural Ireland, with female respondents within their sample cohort (*n* = 215, 40.3%) not reflective of the estimated female population (50.4%) [43]. The authors consider that due to the nature of these studies and the current study which comprise voluntary questionnaires as the primary data collection instrument, potential female respondents may be less likely to participate due to a lack of confidence, self-efficacy or perceived knowledge, thus somewhat mirroring study findings. Accordingly, the authors recommend that future studies seek to address this phenomenon via focused recruitment efforts. 

Female respondents within our study sample exhibited a lower level of awareness of the potential health-risks posed by flooding and were more likely to report a lack of information on water testing as a barrier to getting their water tested. Additionally, women reported lower levels of awareness pertaining to the results of previous groundwater testing, and reported a lower level of information regarding the possible sources of contamination that may exist near their well. Moreover, female respondents indicated that they felt less able to get their water tested following a potentially contaminating flood event; paralleling results by Hynds et al. (2018), in the context of the inspection of onsite domestic wastewater (septic) systems, whereby males were 20% more likely to report being aware of how to inspect their own domestic treatment system [43]. This study finding concerning self-efficacy (one’s belief in one’s ability to succeed in specific situations or accomplish a task) is noteworthy as it has been shown that self-efficacy can have a positive influence or in some cases represent a necessary precursor to preventive behaviours [44]. This may be particularly significant regarding “pro-environmental” and/or healthy behaviours as they refer to the occurrence, movement and contamination of groundwater systems, with these concepts and processes typically the least well understood within the global hydrological cycle [45,46]. Additionally, female respondents from the current study did not believe that water testing post-flood was a normative behaviour (e.g., those behaviours agreed upon by society as being “correct”) within their community. Thus, women reported lower self-efficacy (subjective skill) and industrial knowledge, lower risk awareness and less normative influence than was exhibited by male respondents. However, further statistical analysis using the RANAS measures did not discover any moderation by gender, indicating that the RANAS model’s capacity to predict post-flood behaviour was not conditional upon respondent gender. Two RANAS factors that significantly predicted post-flood behaviour for the group as a whole were, Normative factors (social beliefs that one’s community either promotes or forbids a given behaviour) and Perceived Risk factors (actually recognising and emotionally responding to a given phenomenon); two of the domains in which women and men significantly differed (with Ability being the third). A study by Hynds et al. (2013) sought to assess general levels of risk perception among 245 Irish private well users, and found that measured awareness was low with respect to both the sources of and pathways for microbial contamination of their domestic source; well users are not well versed in the bi-directional interactions between surface water and groundwater [47]. Accordingly, in the absence of mechanistic understanding, risk perception will be concurrently low or absent. Hynds et al. (2013) also reported that lower levels of perception were associated with an increased contraction of (likely) waterborne infection, thus the link between awareness, perception and domestic/public health is not in question [47]. More generally, McCright (2010) has shown that women typically underestimate their “climate change knowledge” more than men, potentially affecting both their levels of perceived risk and ability [48].

In examining the pattern of post-flood behaviour reported within each gender sub-sample, female respondents reported a higher likelihood of undertaking “smaller” actions located within the home, such as boiling water or changing their drinking water source, and were more likely to actively seek out information on water testing. As reported by Cvetković et al. (2018), following significant flooding in Serbia during 2014, women displayed a deeper understanding of flooding events, demonstrating more household-caring attitudes and behaviours [49]. Male respondents within the current study were more likely to engage in increasingly ”physical” (or “male-typical”) behaviours such as using sandbags to prevent contamination before/during flooding and to treat (chlorinate) water prior to consumption. Again, Cvetković et al. (2018) cite similar findings, insofar as men seem to be more confident in their abilities to cope with flooding, perceiving greater individual and household preparedness. The 2014 flooding experienced in Serbia was characterised by poor response management accentuated by a gender imbalance, leading to significantly adverse outcomes [49]. Most sample respondents within the current study were conjecturally amenable to taking post-flood actions, however, patterns of behaviour undoubtedly differ by gender, with respondents observed to fall more readily into their “traditional” areas of control i.e., within and outside of the household [50,51]. Study findings agree with the social and socio-developmental underpinnings of the RANAS model, in that social roles, which are frequently attributed to normative socialisation across lifespans, in addition to biological differences [33], tend to drive the types of behaviours that people perform (i.e., lower levels of risk perception among female respondents may be due to private well knowledge and/or maintenance being perceived as “male-typical”). 

This may also be used to elucidate the role that gender-typical knowledge, which is often transmitted through social interactions with others, plays in informing behaviour. Thus, although the RANAS factors did not appear to be moderated by gender, bivariate and descriptive analyses indicate that gender plays a meaningful, albeit conservative role in post-flood actions, which would seem to agree with previous studies which have documented a small but persistent gender gap in environmental awareness, concern, and perceptions [49]. Study findings suggest that optimisation and implementation of hydrological interventions should present an increasingly wide a range of information and tailor information to particular audiences. For example, the fact that men were less likely to take action if previous testing had not reported contamination indicates a significant misunderstanding of flooding as a dynamic vector for contamination [1]. As such, male audiences should be educated around biological and environmental “contaminant pathways” and the inherently fluid (i.e., temporal) nature of these. Conversely, among female audiences, previous studies have suggested that a targeted approach could help improve self-efficacy of females by increasing their confidence in the ability to protect themselves and their property [40]. Therefore, it is necessary to highlight the risk of groundwater contamination to motivate action, however without providing actionable information about the practical steps that can be taken to mitigate the effects of flooding, this intention is unlikely to result in significant behavioural change. For example, in many regions, routine household tasks are more often performed by females [49,50,51]; however, activities such as repairs and maintenance are more often undertaken by males [52]. Accordingly, while men may exhibit lower levels of flood risk perception, they may be as, or more, likely as women to undertake mitigation behaviours. 

## 5. Conclusions

To the authors’ knowledge, the current study represents the first gender-based investigation of flood risk perception and mitigation behaviours among private groundwater users, a particularly susceptible population with respect to waterborne infection as a result of surface flooding. Findings suggest that gender differences do exist, and as such, may affect human health risk during and immediately after flood events. Female survey respondents exhibited lower levels of awareness of the potential health-risks posed by flooding, while they reported a higher likelihood of undertaking physically reduced actions within the home such as boiling water or changing their drinking water source (e.g., switching to bottled water). Conversely, male respondents were more likely to engage in more ”physical” (or “male-typical”) behaviours such as using sandbags to prevent contamination before/during flooding and treating (chlorinate) water prior to consumption. Thus, “gendered behaviours” might be generally described as being internal (female) and external (male). As such, the authors recommend that public authorities seek to purposefully engage with both male and female private groundwater users to increase household participation and shared responsibility during extreme hydrological events i.e., take advantage of “mixed-gender” physical and non-physical behaviours. Gender-specific or gender-focused methods of communication should be employed where possible (e.g., (wo)men’s interest groups, focused traditional media outlets including magazines, newspapers, radio and television). Moreover, messaging should seek to outline and facilitate the ease with which traditionally gendered behaviours can be undertaken irrespective of gender (e.g., females can easily undertake behaviours outside the home if required and vice versa). While the authors consider that the current study accurately represents the population being studied (i.e., private groundwater users in the Republic of Ireland), further research is recommended to verify the transferability of study findings, and use this as a baseline for future “gendered” (or gender-sensitive) intervention design and message framing. 

## Figures and Tables

**Figure 1 ijerph-17-02072-f001:**
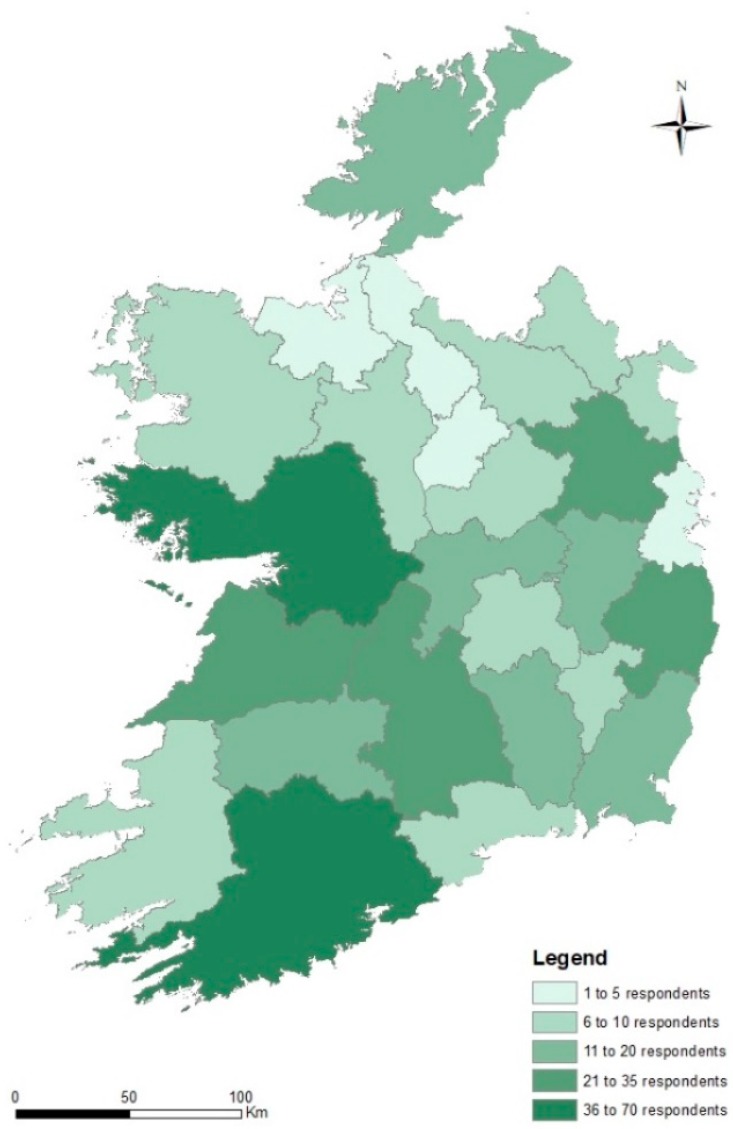
Survey responses by administrative county in the Republic of Ireland. Used with permission from Andrade et al. (2019).

**Table 1 ijerph-17-02072-t001:** Included Risk, Attitude, Norms, Ability, Self-regulation (RANAS) framework factors and their assessment in the survey.

Factor Blocs	Factors	Assessment (All Assessed on a 1–5 Likert Scale)
Risk factors	Perceived Vulnerability	“My well can become contaminated if flooding occurs within 100 m (110 yards) of it”
	Perceived Severity	“My life would be impacted if I or a member of my household became ill with symptoms of diarrhoea and/or vomiting”
	Factual Knowledge	“You can always tell when well water is contaminated by its taste, colour or smell” “Wells can stay contaminated for weeks after the flood period has passed”
Attitude factors	Instrumental Beliefs	“Getting my well water tested in a laboratory is an easy task”
	Affective Beliefs	“After a flood I would worry less knowing that my well water is tested by a laboratory”
Norm factors	Descriptive	“People I know would test their well water if flooding occurred near their well”
	Injunctive	“People who visit me expect me to ensure my well water is safe to drink and not contaminated”
	Personal	“I would feel personally obligated to test my well water after flooding occurred near my well” “If I notice that my well is flooded, I would feel personally obligated to test my well water”
Ability factors	Action Knowledge	“I know who to contact to get my well water tested”
	Self-efficacy	“I am able to get my well water tested if I decide to”
Self-regulation	Commitment	“I will test my well water if flooding occurs nearby”

**Table 2 ijerph-17-02072-t002:** Variables and their assessment in the survey.

Variable Name	Assessment
Gender ^1^	Participant self-reported gender as either male or female
Flood experience	Participant reported whether they had direct personal experience of flooding
Following of EPA ^2^ guidelines	Reported following the Environmental Protection Agency water testing guidelines
Well location awareness	Awareness of well location in relation to possible contamination sources
Testing awareness	Have you ever had your well water tested for contamination?
Test results awareness	Has a test of your well ever shown signs of contamination? Yes/no = aware, don’t know = unaware
Treatment awareness	Do you apply any treatment to your well water before drinking?
Microbial water treatment	Participant reported using chlorination, UV light, reverse osmosis, or boiling to treat their water
Prior water testing experience	Participant reported having tested their well water before
Primary post-flood actions	Participant reported what they believed to be the/most important action post flooding near groundwater supply
Reasons for not testing	Participant reported reasons for not testing their water or doing so more often

^1^ “Prefer not to say” and “Other” excluded from analyses due to excessively low sample sizes. ^2^ Environmental Protection Agency.

**Table 3 ijerph-17-02072-t003:** Gender-related differences within RANAS components derived from ANOVAs and quantified by Hedges’ *d*.

RANAS Factor Blocs	RANAS Factors	Male (*n* = 237) Mean ± SD	Female (*n* = 168) Mean ± SD	*p*-Value	*d* (95%CI)
Risk factors	Perceived Vulnerability	3.38 ± 1.23	3.46 ± 1.22	0.548	0.07 (−0.13 to 0.26)
	Perceived Severity	4.25 ± 0.73	4.35 ± 0.80	0.230	0.13 (−0.07 to 0.33)
	Factual Knowledge	3.86 ± 0.74	3.83 ± 0.44	0.710	−0.05 (−0.24 to 0.15)
Attitude factors	Instrumental Beliefs	3.22 ± 1.16	3.10 ± 1.17	0.314	−0.10 (−0.30 to 0.09)
	Affective Beliefs	4.18 ± 0.84	4.11 ± 0.90	0.396	−0.08 (−0.28 to 0.12)
Norm factors	Descriptive	3.06 ± 0.90	2.88 ± 0.89	**0.045**	−0.20 (−0.40 to 0.00)
	Injunctive	3.87 ± 0.93	3.71 ± 1.07	0.122	−0.16 (−0.36 to 0.04)
	Personal	3.92 ± 0.82	3.84 ± 0.85	0.344	−0.10 (−0.29 to 0.10)
Ability factors	Action Knowledge	3.65 ± 1.28	3.52 ± 1.32	0.335	−0.10 (−0.30 to 0.10)
	Self-efficacy	4.00 ± 0.97	3.78 ± 1.05	**0.030**	−0.22 (−0.42 to -0.02)
Self-regulation	Commitment	3.70 ± 1.00	3.67 ± 0.97	0.781	−0.03 (−0.23 to 0.17)

Bold text indicates significance at the *p* < 0.05 level; 95% CI = 95% confidence interval; RANAS = Risk, Attitude, Norms, Ability, Self-regulation; SD = standard deviation.

**Table 4 ijerph-17-02072-t004:** Gender-related differences in water treatment and testing awareness and post flood behaviours derived from Chi-square tests.

	Male N (%)	Female N (%)	*p*-Value
Flood experience (*n* = 379)			0.20
Direct	44 (11.6)	36 (9.5)	
Indirect	52 (13.7)	24 (6.3)	
None	131 (34.6)	92 (24.3)	
Follow of EPA guidelines (Yes; *n* = 390)	48 (20.8)	28 (17.6)	0.44
Well location awareness (Yes; *n* = 405)	229 (96.6)	151 (89.9)	**0.01**
Testing awareness (Yes; *n* = 405)	231 (97.5)	159 (94.6)	0.14
Test results awareness (Yes; *n* = 299)	183 (98.9)	106 (93.0)	**0.01**
Treatment awareness (Yes; *n* = 405)	233 (98.3)	161 (95.8)	0.13
Microbial water treatment (Yes; *n* = 394)	76 (32.6)	40 (24.8)	**0.01**
Prior water testing experience (Yes; *n* = 390)	185 (80.1)	114 (71.7)	0.05
Primary post-flood actions (*n* = 405)			**0.05**
Seeking information on what to do to ensure well water is safe to drink	44 (18.6)	34 (20.2)	
Boiling well water before consuming it	13 (5.5)	16 (9.5)	
Disinfecting (chlorinating) well water	9 (3.8)	1 (0.6) *	
Testing well water	59 (24.9)	32 (19.0)	
Drinking water from other sources (incl. bottled water)	30 (12.7)	27 (16.1)	
Trying to prevent contamination ingress in well (e.g., sandbagging)	17 (7.2)	9 (5.4)	
Not aware that action should be taken	21 (8.9)	27 (16.5) *	
Take no action	29 (12.2)	16 (9.5)	
Other	15 (6.3)	6 (3.6)	
Reasons for not testing (*n* = 314)			
Lack of information on testing	9 (4.9)	15 (11.5) *	**0.03**
Takes too much time	5 (2.7)	2 (1.5)	
Difficulties in collecting samples	2 (1.1)	0 (0.0)	
Operating lab hours	5 (2.7)	2 (1.5)	
Too far from lab	6 (3.3)	4 (3.1)	
Too expensive to treat if there are problems	5 (2.7)	9 (6.9)	
Cost of water testing	20 (10.9)	23 (17.6)	
Previous test results were normal	48 (26.2)	16 (12.2) *	
Use a water treatment system before drinking it	11 (6.0)	7 (5.3)	
No water quality problems in my local area	15 (8.2)	10 (7.6)	
No reason to be concerned about well water quality	57 (31.1)	43 (32.8)	

Bold text indicates significance at the *p* < 0.05 level; * denotes a subset of each category whose column proportions significantly differ from each other at the *p* < 0.05 level; EPA = Environmental Protection Agency; SD = standard deviation.

**Table 5 ijerph-17-02072-t005:** Odds ratios (OR) and 95% confidence intervals (CI) derived from binominal logistic regression analyses as indicators of associations between RANAS components and experiential health behaviours following flooding among people with direct experience of flooding (*n* = 80).

RANAS Factor Blocs	Behavioural Factors	OR (95% CI) (Per 1-Unit Increase in RANAS Factor)	*p*-Value (RANAS Factor)	*p*-Value (RANAS X Gender Interaction)
Risk factors	Perceived Vulnerability	5.57 (1.90 to 16.31)	**<0.01**	0.36
	Perceived Severity	1.55 (0.42 to 5.72)	0.51	0.68
	Factual Knowledge	2.30 (0.79 to 6.73)	0.13	0.41
Attitude factors	Instrumental Beliefs	1.14 (0.61 to 2.13)	0.67	0.48
	Affective Beliefs	1.09 (0.48 to 2.47)	0.84	0.88
Norm factors	Descriptive	1.73 (0.75 to 3.98)	0.20	0.21
	Injunctive	2.41 (0.96 to 6.04)	0.06	0.86
	Personal	4.38 (1.33 to 14.39)	**0.02**	0.92
Ability factors	Action Knowledge	1.31 (0.78 to 2.21)	0.30	0.51
	Self-efficacy	1.38 (0.74 to 2.57)	0.31	0.90
Self-regulation	Commitment	4.70 (1.70 to 13.02)	**<0.01**	0.19

Bold text indicates significance at the *p* < 0.05 level; 95% CI = 95% confidence interval; RANAS = Risk, Attitude, Norms, Ability, Self-regulation.

**Table 6 ijerph-17-02072-t006:** Odds ratios (OR) and 95% confidence intervals (CI) derived from binominal logistic regression analyses as indicators of associations between RANAS components and conjectural health behaviours following flooding among people with indirect experience of flooding (*n* = 76).

RANAS Factor Blocs	Behavioural Factors	OR (95% CI) (Per 1-Unit Increase in RANAS Factor)	*p* -Value (RANAS Factor)	*p* -Value (RANAS X Gender Interaction)
Risk factors	Perceived Vulnerability	1.00 (1.46 to 2.19)	1.00	0.16
	Perceived Severity	0.85 (0.28 to 2.55)	0.77	0.35
	Factual Knowledge	2.76 (0.92 to 8.30)	0.07	0.17
Attitude factors	Instrumental Beliefs	1.70 (0.81 to 3.54)	0.17	0.74
	Affective Beliefs	1.72 (0.79 to 3.75)	0.17	0.76
Norm factors	Descriptive	3.42 (1.14 to 10.25)	0.03	0.76
	Injunctive	0.48 (0.17 to 1.37)	0.17	0.08
	Personal	7.30 (2.14 to 24.87)	**<0.01**	0.21
Ability factors	Action Knowledge	1.42 (0.78 to 2.57)	0.25	0.29
	Self-efficacy	1.66 (0.65 to 4.24)	0.29	0.54
Self-regulation	Commitment	2.16 (1.02 to 4.57)	**0.04**	0.33

Bold text indicates significance at the *p* < 0.05 level; 95% CI = 95% confidence interval; RANAS = Risk, Attitude, Norms, Ability, Self-regulation.

**Table 7 ijerph-17-02072-t007:** Odds ratios (OR) and 95% confidence intervals (CI) derived from binominal logistic regression analyses as indicators of associations between RANAS components and conjectural health behaviours following flooding among people with no experience of flooding (*n* = 223).

RANAS Factor Blocs	Behavioural Factors	OR (95% CI) (Per 1-Unit Increase in RANAS Factor)	*p* -Value (RANAS Factor)	*p* -Value (RANAS X Gender Interaction)
Risk factors	Vulnerability	1.99 (1.35 to 2.92)	**<0.01**	0.51
	Severity	1.44 (0.84 to 2.45)	0.19	0.62
	Knowledge	2.30 (1.11 to 4.76)	**0.03**	0.97
Attitude factors	Instrumental	1.73 (1.15 to 2.59)	**<0.01**	0.32
	Affective	1.71 (1.04 to 2.79)	**0.03**	0.54
Norm factors	Descriptive	2.26 (1.23 to 4.19)	**<0.01**	0.52
	Injunctive	1.58 (0.95 to 2.61)	0.08	0.36
	Personal	5.47 (2.53 to 11.82)	**<0.01**	0.87
Ability factors	Action knowledge	1.45 (1.02 to 2.04)	**0.04**	0.52
	Self-efficacy	1.60 (1.02 to 2.52)	**0.04**	0.81
Self-regulation	Commitment	4.52 (2.42 to 8.46)	**<0.01**	0.26

Bold text indicates significance at the *p* < 0.05 level; 95% CI = 95% confidence interval; RANAS = Risks-Attitudes-Norms-Abilities-Self-regulation.

**Table 8 ijerph-17-02072-t008:** Gender differences in conjectural and experiential self-reported behaviour within people with direct, indirect, and no flooding experience derived from Chi-square tests.

	Male N (%)	Female N (%)	*p* -value
Direct experience (tested water)	11 (25%)	11 (30.6%)	0.58
Indirect experience (would test water)	44 (84.6%)	21 (87.5%)	0.74
No experience (would test water)	109 (83.2%)	76 (82.6%)	0.91

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
