# Peer review of "Gender-Related Differences in Flood Risk Perception and Behaviours among Private Groundwater Users in the Republic of Ireland"

_ijerph, 2020, doi:10.3390/ijerph17062072_

Round 1
Reviewer 1 Report
This was a joy to read!
Your article compared the risk perceptions of a group of people within Ireland along male/female lines to determine whether there was a difference in post flood behaviours and perceptions. This was an incredibly interesting study that could pave the way in terms of effective risk communication in the future. Particularly interesting was that your findings indicate that females were less likely to report that post-flood action on their wells was needed. This is quite fascinating in the context of much research out there that describes women as being the key communicators in disaster risk planning (due to their effective action planning). In particular, previous research specifically highlighted middle aged women (from memory it was around 45-54) that were most responsive to disaster risk.
That aside, I thought this article is very important in understanding post-flood behaviours and perceptions for private water supplies. It particularly highlights some important measures to take with reference to promoting better responses, and management of on-site systems.
I have a few minor points that you may want to consider:
Lines 30-31: This is an important line, but it is confusing- consider rewording.
Line 48: why compare only to Canada- make this clear. Could it because it has a similar economic standing or for the way their water systems are set up?
59-60: 'myriad' written twice in the same sentence. This whole sentence is value, try to reword to be more specific.
100-102: don't need this here: maybe in acknowledgements or end of paper? See with editor.
109: Participant reported as either male or female. An important element to your paper is to describe why you have not considered other genders in your survey (non-binary incl transgender, intersex etc). Or did you omit the 'other' from your survey? If you did, make clear why you did this- perhaps a paragraph on this, as it does leave out an important segment of society.
145. 58.5% of participants were male: does this closely reflect the gender distribution in Ireland/in the region? If so, mention this.
275-277: "Additionally.... informing behaviour." This is such an incredibly important point! Gets a bit lost in the middle of the paragraph, consider making the point closer to the top of a paragraph if possible.
309: "the authors recommend that public authorities seek to purposefully engage with both male and female..." - should state in what way public authorities should engage. Try to be specific- it's a bit vague as it is.
Ensure references are consistent in style.
Overall, a great read!
Author Response
ijerph-729708 – Response to Reviewer
Gender-related differences in flood risk perception and behaviours among private groundwater users in the Republic of Ireland
Reviewer 1 Comments and Author Responses
Reviewer Comment 1: This was a joy to read!
Author Response 1: Thank you very much! We’re delighted that the reviewer enjoyed the article!
Reviewer Comment 2: Your article compared the risk perceptions of a group of people within Ireland along male/female lines to determine whether there was a difference in post flood behaviours and perceptions. This was an incredibly interesting study that could pave the way in terms of effective risk communication in the future. Particularly interesting was that your findings indicate that females were less likely to report that post-flood action on their wells was needed. This is quite fascinating in the context of much research out there that describes women as being the key communicators in disaster risk planning (due to their effective action planning). In particular, previous research specifically highlighted middle aged women (from memory it was around 45-54) that were most responsive to disaster risk.
Author Response 2: Yes, we were slightly surprised by our findings in light of previous reporting, however, much of the impetus underlying our study was the fact that we have previously found that private well owners do not necessarily adhere to perception- or action-based “norms” with respect to environmental behaviours and psychology in general, likely due to their unregulated status
Reviewer Comment 3: That aside, I thought this article is very important in understanding post-flood behaviours and perceptions for private water supplies. It particularly highlights some important measures to take with reference to promoting better responses, and management of on-site systems.
Author Response 3: Thank you. We hope that readers including policy makers and “non experts” might find this study and related findings useful
I have a few minor points that you may want to consider:
Reviewer Comment 4: Lines 30-31: This is an important line, but it is confusing- consider rewording.
Author Response 4: We agree, this was slightly awkwardly worded; we have tried to clarify as follows:
“Female respondents reported a lower level of awareness of the need for post-flood action(s) (8.9% vs 16.5%), alongside a perceived “lack of information” as a reason for not testing their domestic well (4.9% vs 11.5%).”
Reviewer Comment 5: Line 48: why compare only to Canada- make this clear. Could it because it has a similar economic standing or for the way their water systems are set up?
Author Response 5: From our perspective, Canada and particularly Ontario is a world leader in the field of private groundwater research, and as such, tend to have the most reliable/accurate data with respect to reliance rates, etc. In order to provide a little more balance here, we have included the estimated US figure (48 million, 14.7%) in order to make it clearer that we are simply providing some comparative examples.
Reviewer Comment 6: 59-60: 'myriad' written twice in the same sentence. This whole sentence is value, try to reword to be more specific.
Author Response 6: We agree, this was a rather poorly written sentence. We have rewritten as follows:
“Over the past three decades, the multiple social, economic, demographic, and geographical factors influencing people’s perceptions and responses to risk in numerous domains including personal health and wellbeing, environment, education and finance has received increasing attention [9,10].”
Reviewer Comment 7: 100-102: don't need this here: maybe in acknowledgements or end of paper? See with editor.
Author Response 7: We entirely understand the reviewers perspective, however as there is no acknowledgements section, we have elected to keep this statement here unless the editor disagrees. We feel it is most appropriately situated within the methods section as it represents a necessary component of study (survey) design due to our collecting data from private individuals.
Reviewer Comment 8: 109: Participant reported as either male or female. An important element to your paper is to describe why you have not considered other genders in your survey (non-binary incl transgender, intersex etc). Or did you omit the 'other' from your survey? If you did, make clear why you did this- perhaps a paragraph on this, as it does leave out an important segment of society.
Author Response 8: This is a very pertinent and topical point. From the authors perspective, in the interest of survey brevity, we did not consider it feasible to provide a comprehensive list of non-binary gender classifications in our survey, and in order to ensure analytical comparability across respondents, we had decided during questionnaire development to not include any open response options which would have been necessary in the absence of explicit non-binary genders being provided as response options. While “prefer not to say” and “other” were indeed included as response options, as both of these received <2% of participant responses, we elected (perhaps inappropriately) to exclude these from analyses as this would have required the use of far more complicated statistical approaches than those used (e.g. maximum likelihood estimates in logit regression are known to suffer from small sample bias, thus necessitating “exact models”). As such, we selected to employ simpler modelling and interpretation for the sake of readability. This has now been explicitly stated in the amended manuscript (Table 2, Footnote).
Reviewer Comment 9: 145. 58.5% of participants were male: does this closely reflect the gender distribution in Ireland/in the region? If so, mention this.
Author Response 9: The reviewer makes a very pertinent comment that we should have addressed in the first version of the manuscript. We have sought to clarify this via the following additional text (Situated in discussion):
“It is important to note that according to the most recent Irish census (2016), approximately 50.4% of the national population identifies as female [38], and thus, the current study sample does not strictly reflect the gender distribution within the Republic of Ireland (i.e. female sub-sample, 41.5%). Previous studies of environmental issues within rural areas, and particularly those focusing on water resources and waste management have reported similar difficulties in recruiting gender-balanced survey cohorts [28, 31, 39]. Flanagan et al. (2015) previously surveyed 525 private groundwater-reliant householders across central Maine (US) regarding well testing and treatment practices in an area characterised by high geogenic groundwater arsenic [39]. Although the male/female ratio within the background population was 49.4%/50.6%, the surveyed sample displayed a slight gender skew of 54%/46%. Similarly, Hynds et al. (2018) surveyed 533 septic tank users from rural Ireland, with female respondents within their sample cohort (n = 215, 40.3%) not reflective of the estimated female population (50.4%) [28]. The authors consider that due to the nature of these studies and the current study which comprise voluntary questionnaires as the primary data collection instrument, potential female respondents may be less likely to participate due to a lack of confidence, self-efficacy or perceived knowledge, thus somewhat mirroring study findings. Accordingly, we recommend that future studies seek to address this phenomenon via focused recruitment efforts.”
Reviewer Comment 10: 275-277: "Additionally.... informing behaviour." This is such an incredibly important point! Gets a bit lost in the middle of the paragraph, consider making the point closer to the top of a paragraph if possible.
Author Response 10: The authors agree that the issue of gender-specific knowledge was rather lost within the paragraph; we have elected to rephrase this sentence slightly and move it down so that it opens the final discussion paragraph
Reviewer Comment 11: 309: "the authors recommend that public authorities seek to purposefully engage with both male and female..." - should state in what way public authorities should engage. Try to be specific- it's a bit vague as it is.
Author Response 11: We agree that a little more detail was needed here. Accordingly, and with the fact that this is the conclusion section being kept in mind, we have sought to provide examples of potential/recommended engagement structures, while trying to remain brief, by adding the following statements directly after the statement in question:
“Gender-specific or gender-focused methods of communication should be employed where possible (e.g. (wo)mens interest groups, focused traditional media outlets including magazines, newspapers, radio and television). Moreover, messaging should seek to outline and facilitate the ease with which traditionally gendered behaviours can be undertaken irrespective of gender (e.g. females can easily undertake “internal” behaviours if required and vice versa).”
Reviewer Comment 12: Ensure references are consistent in style.
Author Response 12: We have double-checked all referencing to ensure consistency
Reviewer Comment 13: Overall, a great read!
Author Response 13: Thank you!!
Reviewer 2 Report
The topic of post-flooding groundwater contamination behaviours in a gendered perspective in Ireland has the potential to be relevant (and necessary) for public health authorities and the research arena. However, due to the major issue hereafter highlighted I don’t think it is ready to be published. I recommend the authors to re-think about the rationale of the study (why gender and why Ireland) and the key message they want to deliver (to whom this research is addressed and for what purpose, i.e. recommendations for actions). Some major and minor comments are discussed below.
Major comments:
I suggest to re-conceptualize the introduction because the concepts expressed seem to be detached ones from the others. The authors started with flooding causing contamination (general), then projection over floods (in Ireland), domestic groundwater use (in Ireland) and users’ aware of flooding for their supply (general) (this just within the first 14 lines of the introduction). There is a missing link between the need for increasing awareness and gender (lines 59-60). Maybe it is necessary to add a couple of lines in reference to the importance of public opinions in risk-taking responsibilities before starting from the factors (e.g. gender) influencing it. A reader can be lost after reading the jump from Ireland to gender and again from gender to post-flood risks and mitigation behaviours (line 67 onwards). The literature review (lines 72-84) is very limited and appear to be a mere list of papers without a proper understanding of the meaning of gender in flood risk research.
The justification presented in lines 85-90 is far from being exhaustive. The gap in post-flood gendered perception needs to be justified strongly as well as with the decision to undertake this study in Ireland. The research gap is not well addressed. An Irish history of previous post-flood contaminations of groundwater or some flooding trends in the country needs to be explored as well as the decision to focus only on the gender dimension as well. This because there are many other socio-economic variables that have much more influence in explaining the decision of undertaking protective behaviours, e.g. income, family structure (dependent children), education. The authors need to justify why solely gender has been accounted for the current research. Which is the overall influence that this variable brings in regard to post-flooding contamination information and testing? Why is it important to know if men or women have much concern? I would suggest to add also a more general analysis (with other variables, taken as a single or as an interaction) to understand broadly the studied population, concern and choices over post-flooding groundwater contamination.
The methodology has some limitations due to the nature of the sample choice. How the sample has been controlled in terms of geography, age, education, marital status and or income, house property, etc.? Online surveys bring a certain level of sample selection bias that needs to be further explored. Also, which was/were the hypothesis/hypotheses tested? What is the gender difference expected and what’s the challenge of knowing it regarding post-flooding mitigation choices?
The paper lacks a proper discussion of results in a policy-making point of view. What do these results are useful for? Which are the gendered recommendations that can be suggested/concluded? Why is this research usable, useful and used? The researchers need to contextualise the findings and give insights for practical relevance. Many of the statistical tests are negative. Does it mean that gender is not an influencing (thus not essential) variable in this regard? I suggest thinking whether to change the manuscript into a more general explorative one (including all socio-economic variables) and keep the statistically significant gendered analysis.
Minor comments:
- Please consider to split (or rephrase) the sentence on pg. 1 lines 42-46.
- I cannot understand the comparison with the Canadian population on pg. 2 lines 49-50. Did the authors intend to show the different magnitude (same percentage but over a different total amount) or to list other countries that use private groundwater systems? Maybe the sentence should be a little bit rephrased in this respect (otherwise the inclusion of Canada example comes without justification).
- “Frequently non-regulated nature” sounds a bit odd. I would suggest to re-phrase the entire paragraph (pg. 2 lines 56-59) since it is too complicated to read
- Please the double repetition of “myriad” in pg. 2 lines 60-61. In addition, this sentence is too vague to have a proper meaning. I suggest concentrating on the information relevant for understanding the results.
- What are ‘environmental behaviours’ in lines 67?
- Which are the ‘non-professional interests group’ specified on line 96?
- What is the third level of education?
- Why the online survey grasped the majority of household in rural agricultural settlements and not in major urban city settlements?
- I cannot understand the difference between an indirect experience (would test water) and no experience (would test water) of table 8. Indirect experience is the experience others’ did that can reflect one’s behaviours; is that the case?
- The ‘previous research’ (pg. 10 lines 199-201 and same page lines 205-206) is examined in Ireland?
Author Response
Please see uploaded document for point-by-point responses

Round 2
Reviewer 2 Report
The authors provided a good number of revisions except for the most important one: they were unable to justify why they have chosen gender as the key variable in this research. Being repetitive, there are many other socio-economic variables that have much more influence in explaining the decision of undertaking protective behaviours, e.g. income, family structure (dependent children), education. Using gender, in this context, seems unjustified and under random selection. I think that answering these questions would help the authors to enrich the research gap:
1) Which is the overall influence that this variable brings in regard to post-flooding contamination information and testing?
2)Why is it important to know if men or women have much concern?
The justification provided that "This knowledge gap represents a key impediment to the development of ‘fit for purpose’ risk reduction, as highlighted in the Hyogo Framework for Action, 2005-2015 [19]" is outdated. There are new political schemes in place now that the authors should refer to. In addition, they should narrow down the research gap on Ireland rather than claiming the international political agenda.
In addition, researchers identify Ireland because it is the country of the authors. Please, don't use this justification in the future; it gives less importance to your study. In addition, I strongly suggest to add some information on the flood history of Ireland and on flood-contamination. Please refer to some data and trends, and avoid general statements.
There is a need for a robust rational and robust implications of the research for the community, for policy-makers, practitioners or whoever. For this reason, the implications of this research are not exhaustive to my point of view. This addition “Gender-specific or gender-focused methods of communication should be employed where possible (e.g. (wo)mens interest groups, focused traditional media outlets including magazines, newspapers, radio and television). Moreover, messaging should seek to outline and facilitate the ease with which traditionally gendered behaviours can be undertaken irrespective of gender (e.g. females can easily undertake “internal” behaviours if required and vice versa).” is a very general discussion point of view by adding just the word "gender". Please, read some gender literature and discuss the relevance of this paper in a more detailed way. Avoid to be general, you need to discuss it according to your knowledge of Ireland customs, traditions (even gendered), behaviours and environmental policy. In addition, the sentence in the brackets is a bit odd. Please, conder to rephrase it, since "internal" behaviours can be misinterpreted.
The paper has a lot of potential, I share the authors' point of view regarding its novelty but still misses some important details.
Author Response
Reviewer 2 – Comments to Authors
General Author(s) Response: We thank the Reviewer for their further comments. We have carefully considered all suggestions and have provided further detail and clarity, where appropriate. We believe that the manuscript has been substantially strengthened as a result of these revisions and hope that this is to the satisfaction of the reviewer.
Reviewer comment 1: The authors provided a good number of revisions except for the most important one: they were unable to justify why they have chosen gender as the key variable in this research. Being repetitive, there are many other socio-economic variables that have much more influence in explaining the decision of undertaking protective behaviours, e.g. income, family structure (dependent children), education. Using gender, in this context, seems unjustified and under random selection.
Author(s) Response: We appreciate the opportunity to provide clarity regarding the focus on gender in the current manuscript. We do not wish to assert that gender is the most influential socio-economic/demographic variable in driving the decision to undertake protective behaviours. However, gender has been shown to be an influential factor in risk perception, and thus worthy of investigation. Moreover, the role of gender in flood risk perception and mitigation behaviours among private groundwater supply users has not been explored. Therefore, the current study aimed to address this research gap, both in the Republic of Ireland generally (where flood risk perception has not been previously explored that we can see in the scientific literature), and specifically within a cohort of private well users (Which as best we can ascertain, has not previously done nationally or globally). This is addressed in the below passage (in the original manuscript), but in line with reviewer concerns, we have added further justification (in red) to help support the study rationale.
Lines 82-107: “Historically, gender has been an understudied construct of health and medical research [14] with the issue only recently addressed within literature [15]. As such, it is important that any potential gender differences within an environmental health context are assessed [16] as it has now been noted that gender is a vital component of health research [17]. Indeed, understanding gender issues and informing gender sensitive interventions is of particular relevance if we want to deliver more effective health interventions [18]. Within the context of flooding, gender differences have been reported within the literature, for example, several studies have found that female respondents tend to perceive the risk of floods more acutely than their male counterparts [19,20], and thus, may represent a specific target audience for risk reduction strategies. A study of flood-risk perception in the Republic of Ireland found gender differences in relation to the affective component of flood risk perception i.e. females were more likely to worry about natural hazards than males [21]. However, O’Neill et al. (2016) also showed that elevated risk perception did not translate into higher levels of protective behaviour [21], perhaps highlighting that in spite of increased perception among women, traditional ‘gender roles’ (i.e. men as ‘protectors’) prevail and risk reduction measures are influenced more routinely by male perception. Similarly, the probability of purchasing flood insurance has been reported as being comparable (with male respondents) or lower among female respondents [20, 22]. Conversely, Zaalberg et al. (2009) found no association between gender and the intention to undertake adaptive actions for flood damage minimization [23] thus highlighting the inherent complexity of the issue. Notably, the abovementioned studies focused on “pre-flood” perceptions and behaviours; to date, few studies have explored gender related differences on post-flood risk perceptions and behaviours, and particularly as they relate to human health. This knowledge gap represents a key impediment to the development of ‘fit for purpose’ risk reduction, as highlighted in the Hyogo Framework for Action, 2005-2015 [24] and as included in the post-2015 framework for disaster risk reduction, which calls for a gender perspective to be integrated into all disaster management plans, policies, and decision-making processes.”
It is critically important to note that gender is not a merely a variable that assesses the differences between men and women in the wake of disasters. It is also how living conditions, demographic and economic attributes, behaviour’s and beliefs reflect gender power relations in this context.
Reviewer comment 2: I think that answering these questions would help the authors to enrich the research gap:
- Which is the overall influence that this variable brings in regard to post-flooding contamination information and testing?
Author(s) Response: We understand the reviewers’ perspective and wish to highlight that this question is one of the primary questions our research is attempting to answer. For example, within the abstract, we have specifically stated “Female respondents reported a lower level of awareness of the need for post-flood action(s) (8.9% vs 16.5%), alongside a perceived “lack of information” as a reason for not testing their domestic well (4.9% vs 11.5%). Conversely, male respondents were more likely to report awareness of their well location in relation to possible contamination sources (96.6% vs 89.9%) and awareness of previous water testing results (98.9% vs 93.0%).” Moreover, the influence of gender on post-flood contamination testing is addressed in the current study with findings demonstrating that following flooding, 30.6% of females and 25.0% of males tested their water for contamination; however, this difference was not statistically significant (p=0.58). Likewise, we have specifically alluded to this issue throughout the article discussion, for example “Additionally, female respondents from the current study did not believe that water testing post-flood was a normative behaviour (e.g. those behaviours agreed upon by society as being “correct”) within their community. Thus, women reported lower self-efficacy (subjective skill) and industrial knowledge, lower risk awareness and less normative influence than exhibited by male respondents.”
- Why is it important to know if men or women have much concern?
Author(s) Response: As above, we believe that we have specifically spoken to this issue throughout the manuscript as it represents the driving objective of the research as a whole i.e. it is critical to understand if ≈50% of any population (in this case private well users, which may comprise up to 40% of rural populations in some regions/nations) thinks, responds, perceives and/or or behaves differently to any extreme climate event (e.g. flooding), and if so, why and how best should these differences be used or mitigated. The implications of understanding gender differences in this area are detailed throughout the manuscript, as are the specific implications of the findings of the current study, for example, the following excerpt details how these differences may be overcome/addressed based on our findings:
Lines 358-367: “Thus, “gendered behaviours” might be generally described as being internal (female) and external (male). As such, the authors recommend that public authorities seek to purposefully engage with both male and female private groundwater users to increase household participation and shared responsibility during extreme hydrological events i.e. take advantage of “mixed-gender” physical and non-physical behaviours. Gender-specific or gender-focused methods of communication should be employed where possible (e.g. (wo)mens interest groups, focused traditional media outlets including magazines, newspapers, radio and television). Moreover, messaging should seek to outline and facilitate the ease with which traditionally gendered behaviours can be undertaken irrespective of gender”
Reviewer Comment 3: The justification provided that "This knowledge gap represents a key impediment to the development of ‘fit for purpose’ risk reduction, as highlighted in the Hyogo Framework for Action, 2005-2015 [19]" is outdated. There are new political schemes in place now that the authors should refer to. In addition, they should narrow down the research gap on Ireland rather than claiming the international political agenda.
Author(s) Response: We appreciate the opportunity to strengthen the rationale for the current study. Presently, there are no political documentation, regulations, strategies, communication plans, policies and/or formal guidance available in Ireland which specifically refer to gender within the context of flood management, mitigation, or planning. Moreover, as an EU Member State, and due to the fact that increasing flood severity, incidence and intensity is a global issue, we have chosen to frame this issue within a global context. Likewise, myriad studies have shown that private groundwater users are a susceptible population irrespective of geographical location. As such, and because the IJERPH represents a truly international journal with an international audience, we chose to stand by our utilisation of international context. Conversely, we agree with the reviewer, that there are more temporally relevant documents/references that could have been specifically referred to.
We have significantly updated this section, as follows:
“…as highlighted in the Hyogo Framework for Action, 2005-2015 [24] and as included in the post-2015 framework for disaster risk reduction”. Moreover, as set out in the Sendai Framework for Disaster Risk Management 2015-2030, on which the Republic of Ireland is a signatory, “Disaster risk reduction requires an all-of-society engagement and partnership. It also requires empowerment and inclusive, accessible and non-discriminatory participation, paying special attention to people disproportionately affected by disasters . . . A gender . . . perspective should be integrated in all policies and practices, and female leadership should be promoted (United Nation Office for Disaster Risk Reduction, 2015). Similarly, World Health Organisation distributed questionnaire among EU Member States found that while most Member States had flood management plans in place, these did not generally address the needs of vulnerable groups or gender considerations (WHO, 2013).
Reviewer Comment 4: In addition, researchers identify Ireland because it is the country of the authors. Please, don't use this justification in the future; it gives less importance to your study. In addition, I strongly suggest to add some information on the flood history of Ireland and on flood-contamination. Please refer to some data and trends, and avoid general statements.
Author(s) Response: We appreciate the reviewers’ perspective, but it is not clear whether s(he) takes issue with Ireland (small country) or any geo-specific case study. Regardless, we would contest that research based within a research teams’ country of origin does not diminish its importance. Rather, the familiarity with social norms, customs, behaviours etc. as well as climate and infrastructure allows for greater clarity when interpreting research findings. Moreover, Ireland is extremely relevant, due to the increasingly frequent incidence of major flood events over the past 2 decades, and our significant reliance on private (unregulated) groundwater supplies in rural areas, thus making it a highly pertinent case study. This has been further outlined in the below passage and information regarding flood history and flood-contamination has been added where available.
Lines 107-122: “The Republic of Ireland, which is characterised by an historic risk of flooding in concurrence with a relatively high level of private (unregulated) groundwater reliance (estimated at over 16% of the population) [6-7], serves as an optimal experimental site for the current study. From the mid-19th century, public policies concerning flooding as it relates to drainage for land improvement for agriculture have been introduced; urban flood events in the 1980s and 1990s saw a policy shift with a focus on protecting urban conurbations and necessary infrastructure from flooding. More recently, there has been a return to wider river-basin concerns and implementation of risk-based models to manage flood risks [25]. However, whilst increasingly holistic risk-based models are being pursued, there has been limited consideration of the link between flooding, contamination and human health [1]. This is in spite of Irish private household wells being identified as the likely source of serious health issues such as verotoxigenic Escherichia coli (VTEC) infections, for which the RoI has the highest incidence rates in Europe [26, 27]. Moreover, recent flooding events have had extensive negative effects [28] with recent regional climate change projections predicting the scenario to worsen in the next 40 years [29, 30]. As such, the Republic of Ireland is a highly pertinent case study to assess the gender-related differences in flood risk perceptions and post-flood mitigation behaviours among private groundwater supply users and is thus, the focus of this study.”
Reviewer Comment 5: There is a need for a robust rational and robust implications of the research for the community, for policy-makers, practitioners or whoever. For this reason, the implications of this research are not exhaustive to my point of view. This addition “Gender-specific or gender-focused methods of communication should be employed where possible (e.g. (wo)mens interest groups, focused traditional media outlets including magazines, newspapers, radio and television). Moreover, messaging should seek to outline and facilitate the ease with which traditionally gendered behaviours can be undertaken irrespective of gender (e.g. females can easily undertake “internal” behaviours if required and vice versa).” is a very general discussion point of view by adding just the word "gender".
Author(s) Response: We appreciate the Reviewer’s suggestion. We have attempted to make general recommendations based on the current findings; however, designing specific interventions is beyond the scope of the current study. In effect, the current study aimed to examine the presence (or absence) of gender-related differences and shine some light on the likely/potential underlying reasons for its presence (if present). Ongoing and future mixed-methods research (focus groups, face-to-face interviews, etc) will draw on these findings and suggestions to design context-specific interventions. However, in the interests of brevity and focus, we don’t believe that these research elements are appropriate in the current manuscript (i.e. integrating multiple research approaches from varying cohorts may introduce confusion and/or would not allow us the space to give both “sides” the space required). This is highlighted in the below passage.
Lines 358-364: “Thus, “gendered behaviours” might be generally described as being internal (female) and external (male). As such, the authors recommend that public authorities seek to purposefully engage with both male and female private groundwater users to increase household participation and shared responsibility during extreme hydrological events i.e. take advantage of “mixed-gender” physical and non-physical behaviours.”
Reviewer Comment 6: Please, read some gender literature and discuss the relevance of this paper in a more detailed way. Avoid to be general, you need to discuss it according to your knowledge of Ireland customs, traditions (even gendered), behaviours and environmental policy.
Author(s) Response: We appreciate the Reviewer’s suggestion; however, as IJERPH is an international journal we do not feel it would be appropriate for the journal’s readership to focus the manuscript too heavily on Irelands customs, traditions or behaviours. As previously mentioned, there is no specific Irish policy on flood risk management as it pertains to gender. Moreover, Ireland is a small country and geo-specific research in this area is relatively limited. To the very best of our knowledge, all relevant Irish studies have already been included and discussed (i.e. surveys and interviews with Irish private well owners/users). Moreover, we feel that we have included a very high proportion of the available literature on flood-risk perception as it pertains to gender. We are mindful of the fact that the discussion section is already approximately 1,600 words in length, and we have already included references to 52 published reports and articles. We hope this meets with the reviewers’ satisfaction.
Reviewer Comment 7: In addition, the sentence in the brackets is a bit odd. Please, consider to rephrase it, since "internal" behaviours can be misinterpreted.
Author(s) Response: We agree, this statement could easily be misinterpreted, with “internal” not being adequately defined. We have revised this sentence as shown below.
Line 367: “(e.g. females can easily undertake behaviours outside of the home if required and vice versa).”
Reviewer Comment 8: The paper has a lot of potential, I share the authors' point of view regarding its novelty but still misses some important details.
Author(s) Response: We appreciate the thoughtful comment and hope the current revisions are to the reviewer’s satisfaction.